# Facilitators and barriers to colorectal cancer screening using the immunochemical faecal occult blood test among an average-risk population in semi-rural Malaysia: A qualitative study

Kogila Ramanathan[1ᵒ], Désirée Schliemann[2ᵒ], Nor Saleha Binti Ibrahim Tamin[3], Devi Mohan[1], Michael Donnelly[2‡]*, Tin Tin Su[1,2,4‡]*

1 Global Public Health, Jeffrey Cheah School of Medicine and Health Sciences, Monash University Malaysia, Selangor, Malaysia, 2 Centre for Public Health and UKCRC Centre of Excellence for Public Health, Queen's University Belfast, Belfast, United Kingdom, 3 Non-communicable Disease—Cancer Unit, Ministry of Health Malaysia, Putrajaya, Malaysia, 4 South East Asia Community Observatory (SEACO), Jeffrey Cheah School of Medicine and Health Sciences, Monash University Malaysia, Subang Jaya, Malaysia

ᵒ These authors contributed equally to this work.
‡ These authors are joint senior authors on this work. MD and TTS also contributed equally to this work.
* michael.donnelly@qub.ac.uk (MD); tintin.su@monash.edu (TTS)

## Abstract

### Background

Colorectal cancer (CRC) incidence in Malaysia is increasing, and most CRC patients are diagnosed at a late stage. This study investigated participant awareness of CRC and their perceptions and views about CRC screening, barriers, benefits, and facilitators towards CRC screening participation as well as health-seeking behaviour and the use of preventative health services.

### Method

Eleven focus group discussions (FGDs) were conducted with a purposive sample of 89 participants aged > 50 from the major ethnic groups in the Segamat District, Johor State. FGDs were audiotaped, transcribed verbatim, and translated into English. Data were analysed using thematic analysis.

### Results

We identified trust in doctors as a key reason for whether or not to seek health care. Generally, the participants had low awareness of CRC sign/symptoms and screening. Emotional and logistic concerns about sending a stool sample to a clinic emerged as the main barriers to screening. Simplified illustrated instructions about stool collection in Malay, Chinese and Tamil, free screening at health clinics and reminders to complete the iFOBT test were perceived to facilitate engagement in screening, and posited as strategies that were likely to increase iFOBT uptake.

**Data Availability Statement:** All relevant data are within the paper and its Supporting Information files.

**Funding:** This study was supported by the Medical Research Council (UK) Global Challenges Research Fund (Ref: MR/S014349/1). The author (KR) received the Graduate Research Merit Scholarship to pursue a Doctor of Philosophy from Monash University Malaysia during the study period.

**Competing interests:** The authors have declared that no competing interests exist.

## Conclusion

Primary care physicians play a crucial role in terms of reducing patient's misperceptions, recommending screening to patients, enhancing attendance, and improving uptake of CRC screening. There is a need for further research to investigate ways in which to reduce identified barriers and implement and test potential facilitative strategies as well as examine adherence by doctors to clinical guidelines about CRC screening.

## Introduction

Globally, colorectal cancer (CRC) is the third most commonly diagnosed cancer and the second leading cause of cancer mortality [1]. CRC cases are expected to increase by 80% by 2035 to 2.4 million and contribute to 1.3 million deaths worldwide [2]. In Malaysia, CRC is the second most common cancer accounting for 13.6% of all cancers (1 in 55 males and 1 in 76 females are at risk of developing CRC in their lifetime) [3]. Preventive screening is essential to detect CRC early and reduce its incidence and mortality [4]. It has been estimated that the proportion of people in Malaysia with a CRC diagnosis who would be alive at least five years later would be approximately 75% if CRC cases were detected early, at Stage I or II, compared to between 17% and 55% if CRC cases were detected at the advanced Stages III or IV [5]. A consensus statement by the Asia Pacific Working Group for CRC stated that CRC screening should be a national health priority due to the rapid increase of CRC cases in Asian countries, including Malaysia [6]. Stool-based tests, i.e. the guaiac Faecal Occult Blood Test (gFOBT) and the immunochemical Faecal Occult Blood Test (iFOBT), are the recommended tests for CRC screening amongst asymptomatic individuals [6] and it has been suggested that iFOBT screening might reduce CRC mortality by up to 70% [7].

Asian countries have employed different strategies to implement and deliver CRC screening. In Malaysia, individuals aged 50–75 years are recommended to be screened every two years with an iFOBT at a health clinic [8]. The uptake of CRC screening has been low, and in 2018, less than 1% of the target population, or around 5.6 million people, were screened for CRC by the Ministry of Health [9]. However, according to the National Health Morbidity Survey (NHMS) study conducted by Ministry of Health Malaysia in 2019, the national coverage for CRC screening using iFOBT was 10.8% [10]. This indicates that some people did their screening at private facilities or Non-governmental organisations (NGOs). A significant increase in iFOBT uptake is required to down-stage CRC. Current opportunistic iFOBT screening reaches only eligible individuals who have a certain age and risk profile who attend government primary care clinics for other health-related issues and to whom a doctor recommends screening. Patients receive a stool container and are asked to return a stool sample to the health clinic for analysis at the laboratory. Patients who receive a positive result are referred for colonoscopy at the nearest public hospital. Thus, there is no population-based screening and uptake of iFOBT screening remains low. There is a need to understand the health screening behaviours of eligible at-risk Malaysians in order to tailor CRC screening programmes according to the requirements of ethnic groups and regions. This study investigated participant awareness of CRC and their perceptions and views about CRC screening, barriers, benefits, and facilitators towards CRC screening participation as well as health-seeking behaviour and the use of preventative health services.

## Methods

Ethical approval for this qualitative study was granted by the Monash University Human Research Ethics Committee (MUHREC: Project ID 20441) and Medical Review & Ethics

Committee (MREC: NMRR- 19-3349-51630), Ministry of Health Malaysia. Written informed consent was obtained from each participant. The Consolidated Criteria for Reporting Qualitative Studies (COREQ) checklist guided the conduct and reporting of the focus group discussions (FGDs), findings, analysis and interpretations [11]. The FGDs were conducted from October to November 2019.

## Study participants

The study was conducted in Segamat, Johor State, Malaysia, as part of a collaborative research programme between the South-East Asia Community Observatory (SEACO) and Queen's University Belfast. SEACO conducts annual surveys in the 5/11 sub-districts of Segamat to the health and wellbeing of semi-urban and rural communities' population health and well-being. Segamat has an ethnic breakdown that is similar to the population profile of Malaysia: Malays (60%), Chinese (23%) and Indians (7%), and a similar gender composition (male: 49%; female: 51%) [12]. The number of adults aged 55–75 years (approximately the target age range for CRC screening) is double that of the national average [13].

The SEACO community engagement committee (CEC) assisted the study team in purposely recruiting male and female participants aged 50 to 75 from the diverse ethnic groups. Participants were required to be fluent in Malay, Tamil or Mandarin. The exclusion criteria were those aged below 50 years of age, participants who were not able to respond independently or who had a history of CRC. A community engagement committee (CEC) is responsible for connecting and engaging residents from the five sub-districts of Segamat in their community and encouraging them to participate in SEACO research activities. The CEC undertakes discussions and activities with local communities and recommends opportunities to enhance community relations and inclusiveness.

FGDs were conducted by a trained female researcher (KR, *Msc*) separately for each gender and ethnic group. FGDs lasted between 60 and 90 minutes and comprised 6 to 10 participants. FGDs were conducted in the language of each ethnicity. The FGDs moderator spoke Malay and Tamil, and a professional Chinese translator co-facilitated and translated the FGDs with Chinese participants. Socio-demographic information was obtained from participants before the start of the FGDs. All FGDs were recorded with permission. FGD participants received a token of appreciation at the end of the session.

## Focus group discussion topic guide

The development of the semi-structured topic guide for the FGDs was guided by the Health Belief Model, relevant literature [14–16] and discussions with two public health physicians, a health psychologist and the wider research team. The interview topic guide consisted of five sections. The first section explored the use of preventive health services among participants. The second section asked about awareness of cancer and screening, in general. The third section explored awareness about, in particular, CRC and CRC screening. In the fourth section, the interviewer explained the procedure for undertaking an iFOBT and then asked questions about the CRC screening invitation and procedure. The final section comprised a brief explanation about current opportunistic CRC screening guidelines in Malaysia and population-based CRC screening programmes in other countries; and then participants were asked questions about barriers and facilitating factors towards CRC screening, as well as preferences for receiving screening reminders and test results. The FGD topic guide was piloted with twelve participants recruited at a SEACO community engagement meeting in Segamat. The pilot FGD comprised male and female participants from the five sub-districts and different ethnicities. The topic guide was reworded and restructured for clarity based on the feedback from the

FGD (S1 Appendix). Female pilot participants reported that women would feel more comfortable being interviewed with other female participants only and separately from male FG participants. As a result, all FG interviews were conducted with males and females separately. Male and female participants were comfortable with the female FG facilitator and researcher.

## Data analysis

The audio recordings were transcribed verbatim and translated into English. Any identifiable information (such as a name) was removed from the transcripts. The transcripts were then verified for their accuracy by the interviewer who listened to the tapes in 15-minute intervals. Hand-written notes from each session were used to supplement the transcripts. Two researchers (KR and DS) independently analysed the FGD transcripts in NVivo 12.0. Data were analysed inductively, and the development of themes involved the systematic search for patterns following a thematic six-phase approach [17]. Firstly, researchers read the data set in-depth at least twice to familiarise themselves with the data. In phase 2, initial codes were generated by systematically identifying and labelling data. Next, both researchers clustered together codes and mapped key themes that emerged from the data. In phase 4, researchers reviewed the fit of the themes with the coded data and ensured each theme had a clear concept. In phase 5, one researcher (KR) then wrote a summary of each theme and, lastly wrote up the results. The research team discussed the interpretation of the data to iteratively refine the analysis and improve the rigour of the research. Originally we planned two FGDs with each ethnic group. However, we conducted three more FGDs with selected ethnicities to ensure data saturation because the number of participants was below six for some of the FGDs. By the third FGD, it was clear that participants faced similar challenges and which was confirmed by the analysis.

## Results

Eleven FGDs comprised a total of 89 participants: Malays (n = 30), Indians (n = 22), Chinese (n = 19) and indigenous residents (n = 18). Table 1 indicates that almost half of the participants were aged between 60 and 69 years, and most (80%) completed primary or secondary education. More than half of the participants (63%) were retired, not working or homemakers.

## Themes

The analysis of the FGD data revolved around five key themes, which appeared to be present consistently across gender and ethnic groups (Table 2). Key- and sub-themes are illustrated and supported with verbatim quotes from participants here. A summary of additional quotes is included in the S2 Appendix.

**Theme 1: Health-seeking behaviour.** Participants reported that they trusted their doctors, followed their advice and did not ask about screening unless it was recommended.

*'We must accept what the doctor is saying and asking us to do. Doctors usually will tell us something which is for our benefit. Therefore, we have to follow the doctor's advice on doing the test.' (Indian-Male)*

However, female participants reported that they were willing to be screened for breast or cervical cancer screening without any signs and symptoms.

*'There was a talk at that hospital, and they offered to do a screening test. I did the mammogram voluntarily.' (Malay-Female)*

**Table 1. Participant characteristics.**

| Characteristics | Gender | | | |
|---|---|---|---|---|
| | Male | | Female | |
| | n | % | n | % |
| *Age range* | | | | |
| 50–59 years | 15 | 34.9 | 20 | 43.4 |
| 60–69 years | 20 | 46.5 | 21 | 45.7 |
| 70–79 years | 8 | 18.6 | 5 | 10.9 |
| *Ethnicity* | | | | |
| Malay | 9 | 20.9 | 21 | 45.6 |
| Chinese | 10 | 23.3 | 9 | 19.6 |
| Indian | 14 | 32.5 | 8 | 17.4 |
| Indigenous | 10 | 23.3 | 8 | 17.4 |
| *Education* | | | | |
| No formal education | 2 | 4.7 | 8 | 17.3 |
| Primary education | 21 | 48.8 | 16 | 34.8 |
| Secondary education | 18 | 41.8 | 20 | 43.5 |
| Tertiary education | 2 | 4.7 | 2 | 4.4 |
| *Employment status* | | | | |
| Pensioners | 8 | 18.6 | 0 | 0 |
| Homemaker/ Not working | 11 | 25.6 | 37 | 80.4 |
| Paid-employee | 16 | 37.2 | 7 | 15.2 |
| Self-employed | 8 | 18.6 | 2 | 4.4 |

Participants who visited their doctor regularly to manage their health conditions (e.g. diabetes or hypertension) stated that they did not ask for other preventative health screening tests when they attended a follow-up appointment (i.e. for a blood test or to receive prescription medication). Chinese and Indian men described having family members who suffered from cancer or other chronic illnesses and how this experience encouraged them to complete an annual medical check-up at a private hospital as a preventative health measure.

*'I used to go to government clinics. I will go every two months once to collect medications for cholesterol and diabetes. I will do a blood test when the doctor asked.' (Indian-male)*

Participants also self-medicated for minor illnesses and went to clinics or hospitals when symptoms worsened. For instance, participants visited a clinic if symptoms, such as pain, persisted after medicating at home. Participants were also comfortable purchasing health supplements and non-prescription medication.

*'It is like this; depending on the symptoms, I will take a health supplement first, see how it is getting on and then go to the nearest clinic if symptoms get worse.' (Chinese-Female)*

Some Indigenous and Malay participants mistrusted modern medicine and preferred to use traditional medicine. Moreover, women were reluctant to go to a clinic when they experienced any health issues and, instead, used traditional oils or herbs. Chinese and Indian participants did not report using traditional medicine.

*'Once, I had a fever. So, I consume tree roots because I do not like manufactured medicines.' (Indigenous-Male)*

**Table 2. Key-themes and sub-themes.**

| Key-themes | Sub-themes | |
|---|---|---|
| Health-seeking behaviours | • Trust in doctor<br>• Participation in secondary prevention, monitoring and management of health conditions<br>• Self-medication for minor illnesses | |
| Cancer Awareness | • Low awareness about CRC signs/ symptoms and screening<br>• Family/ friends with a CRC diagnosis as a source of acquired awareness<br>• Perceived severity of late-stage cancers (general) | |
| Motivation to participate in iFOBT screening | • The presence of self-efficacy<br>• A perception of the benefits of CRC screening and early detection is required to motivate participation in screening and stool sampling. | |
| Barriers to undertaking CRC screening | The absence of an invitation from a doctor to undertake to screen and complete the iFOBT test. | |
| | Barriers to screening (at a clinic) | • Waiting times<br>• Transportation |
| | Financial concerns | • Cost of treatment and medications at private hospitals or clinics |
| | Emotional barriers | • Embarrassment<br>• Disgust |
| | Concerns about posting stool sample | • Unreliability of postal services<br>• Distrust of postal service<br>• Illiteracy and language barrier |
| Suggested strategies to facilitate iFOBT uptake/ completion | • Collaboration with community support groups and non-governmental organisations (NGOs)<br>• Collect stool containers from clinics<br>• iFOBT invitation letters and information leaflets<br>• Returning stool container<br>• Reminders<br>• Receiving results<br>• Health and screening events<br>• Incentives | |

**Theme 2: Cancer awareness.** Participants reported a lack of knowledge about CRC signs, symptoms and screening, including iFOBT and colonoscopy screening. Participants who had previous experience of CRC screening have better awareness about the signs/ symptoms and screening than participants without screening experiences.

> 'I have heard of it but don't quite understand this colon cancer.' (Malay-Female)

> 'Yes, I have done it once. I went to this medical camp and did cancer screening tests, including stool test for colon cancer. They said no problem and I will repeat the test as recommended.' (Indian-Male)

Participants who knew someone with a history of CRC appeared to have a better awareness about CRC symptoms than participants who did not know a CRC patient or survivor.

> 'My friend did get colorectal cancer. It was already a late stage when they learned about it. When they checked, the doctor detected bloody stool and constipation. Then, the doctor advised her to undergo surgery to remove the intestine.' (Malay-Female)

A close relative with a history of CRC appeared to act as a driver to undertake screening.

*'When someone has a family member or friend who experiences cancer, they will be interested in preventative measures, including screening for cancer. Otherwise, people will not bother much about it.' (Indian-Male)*

Participants who had heard about late-stage cancer complications, mainly from a family member or friend with cancer, were aware of the importance of early cancer detection. Participants seemed to understand the importance of early detection of cancer in general.

*'Yes, yes. we knew about cancer if late, no chance, one month (after diagnosis) then it is done.' (Chinese-Male)*

**Theme 3: Motivation to participate in iFOBT screening.** Participants perceived iFOBT screening to be simple and quick. Men appeared to demonstrate a form of self-efficacy to participate in CRC screening in terms of an expressed readiness to collect a stool sample and not reporting feelings of unease or disgust about the procedure. Female participants expressed emotional barriers about participating in iFOBT screening (discussed under Theme 4, below). However, overall, there was a willingness to participate in CRC screening and a recognition that the benefits of early detection outweighed the barriers.

*'I think this test is straightforward to do. It is better to do it to take care of your health. Moreover, this test is free. This is something good for us.' (Chinese-Male)*

**Theme 4: Barriers to CRC screening.** The absence of an invitation from a doctor to undertake screening was one of the barriers to completing CRC screening.

*'Yeah, we do general tests like urine and blood tests at clinics. There was no opportunity to do a cancer screening test like this.' (Indian-Male)*

Malay and Indigenous participants felt that long waiting times and lack of transport were barriers towards screening at public health clinics. Long waiting times at clinics was noted as a barrier for working participants. Older participants relied on their adult children to drive and accompany them to the clinic or hospital for an appointment.

*'I would say here in our sub-district health clinic that it is not a half-hour wait if you want to do anything like that. It is 2–3 hours of waiting. That place is small. Even to pick up medication for fever, it takes 1–2 hours of waiting for your turn.' (Indigenous-Male)*

Financial barriers appeared to be present for private health care users only and were related to the cost of visiting a doctor and receiving treatment at a private clinic or hospital. Public healthcare facilities provide highly subsidised care for all citizens and accessible healthcare services for senior citizens and pensioners.

*'It is better for me to go to the government hospital as financially it is not good to go to a private hospital.' (Indian-Female)*

Female participants voiced feelings of embarrassment and disgust regarding the stool sample provision and collection including a reluctance to handle a stool sample and feeling

unmotivated to complete an iFOBT. In particular, the thought that other people might know that they were carrying a stool sample in a container embarrassed them to the point of feeling even shame.

*'The problem is that it feels disgusting, that is all. That is the problem.' (Malay-Female)*

The postal service was perceived to be unreliable and speed of delivery tended to be slow in rural areas. So, participants were not in favour of the idea of using the postal service to receive stool containers or send samples.

*'Because our postal service is slow.' (Malay-Male)*

Illiteracy was another concern and potential barrier to screening. Indian females and Indigenous participants reported concern about being able to read and understand an invitation letter or information leaflet.

*'We cannot read in Malay. Even in Tamil, it would be difficult if we read and could not understand them. Those who cannot read well would be tough to comprehend the leaflet.' (Indian-Female)*

**Theme 5: Suggested strategies to enhance iFOBT uptake or completion.** Participants suggested several methods to improve iFOBT screening uptake in their communities, such as collaborating with community groups to organise screening events or distributing and collecting stool containers.

*'Through a committee with people familiar with KOSPEN, we will create a committee to execute this programme.' (Malay-Male)*

The Ministry of Health, Malaysia introduced "Komuniti Sihat, Perkasa Negara" (KOSPEN) or translated meaning "Healthy Communities, Building the Nation", in July 2013. This programme aims to reduce risk factors for non-communicable diseases through trained community health volunteers. These trained volunteers function as health agents of change who introduce and facilitate healthy living practices within their respective communities [18].

*'An organisation like Lion Club and KRT has also organised cancer screening tests such as mammograms. Everyone will come.' (Chinese -Female)*

The government established the National Unity and Integration Department and introduced the Neighbourhood Watch *(Kawan Rukun Tetangga -KRT)*. The KRT health plan aims to emphasise community promotion of a healthy lifestyle among neighbours in collaboration with the Ministry of Health [19].

Participants favoured a procedure whereby a stool container is collected from a health clinics after receipt of an invitation letter. Chinese and Indian participants suggested holding dedicated health screening events during which participants would receive containers for samples.

*'There should be an awareness talk about colorectal cancer stool collection bottles can be handled out during the event.' (Indian-Male)*

Overall, there appeared to be a preference for returning a stool sample to a clinics independently.

*'Yes, we will return the specimen bottle to the clinic.' (Indian-Female)*

Regarding views about an intervention to increase screening uptake and the use of a postal intervention, in particular, receiving information about an intervention in advance was perceived likely to improve trust. Chinese, Indian and Indigenous participants reported that the letter and simple information leaflet needed to be produced in three languages, i.e., Malay, Mandarin and Tamil.

*'It would be best to have pictures and three languages. We would not understand if you sent it in English; we would not understand; Malay is still a little more understandable.' (Chinese-Male)*

Participants except Chinese FG members reported the need to receive a reminder such as a phone call from a nurse.

*'Remind us they have our phone number, so call us back. Yes. If you want 100% results, that needs a reminder.' (Malay-Male)*

*'No need as we will be responsible for doing it ourselves. If there is a reminder, you will feel more pressure.' (Chinese-Female)*

Participants preferred to receive results from a doctor (ideally during a face-to-face consultation) if an iFOBT result was positive and a negative result could be communicated through a phone call.

*'It is better to call those who have positive results and ask to come to see the doctor. This would be a practical approach that can be done. It is impossible to call all the patients to go and see the doctor. There are not enough doctors in the clinics. So, this is something to think about.' (Indian-Male)*

Malay and Indigenous male participants suggested that incentives (eg food, presents or free transport) might encourage people to participate in a screening event.

*'Having a campaign and a talk would be good enough. Nevertheless, you should provide a hamper for the participants when organising such events. Just to attract them to the event. It is challenging to make them participate without anything in return.' (Indigenous-Male)*

*'For example, you pay an 'x' amount per trip, RM20. Perhaps, I think many will go. Transport cost is a burden.' (Malay-Male)*

## Discussion

This study explored barriers to, and facilitators of, CRC screening in semi-rural Malaysia. The most commonly reported barriers to screening at clinics were long waiting times and a lack of transport. Lengthy waiting times at health clinics tended to be due to a high number of patients and staff shortages [20]. Suggestions from FGD participants to improve access to CRC screening involved a personalised invitation letter and stool container collection from the clinic. A

desk dedicated to the task of collecting stool containers was suggested as a way to improve the queuing system for walk-in patients. Participants favoured returning their stool samples to the health clinics. Moreover, participants with work commitments suggested a bulk collection due to the hassle of storing a stool sample. Transport was a barrier for participants depending on their family members (predominantly adult children) to drive them to the clinic and hospital appointments. Similarly, a study conducted in Northern Malaysia reported transportation as a barrier to attending health appointments [21].

Emotional barriers (i.e. embarrassment and disgust) were predominately reported by women who were concerned about sampling faeces and described stools as a 'dirty part of the body'. This emotion is a common barrier to iFOBT completion [22,23]. About 30% of the average-risk individuals attending primary care clinics have previously reported fear of a cancer diagnosis, which was associated with more significant avoidance of CRC screening [21]. Men regarded the iFOBT as a convenient, pain-free and simple test. They also demonstrated higher self-efficacy towards attending CRC screening and collecting a stool sample than women. Previous experience of completing an iFOBT or other cancer screening tests positively influenced their attitudes towards subsequent CRC screening.

This study is the first to have explored the perceived opportunities and challenges of delivering stool collection containers via postal services in Malaysia. Studies have suggested that a postal intervention combined with screening text messages and phone calls might increase the uptake of population-based CRC screening [24–26]. However, participants in this study described several potential challenges, such as distrust in, and the inefficiency of, postal services and literacy issues in communities. There is a need to increase accessibility and improve communication regarding CRC screening interventions in Malaysia by providing, for example, simple instructions written in local languages (Malay, Mandarin and Tamil) with clear illustrations and possibly, instructive videos. A study among rural older adults found that mail-delivered printed educational materials and a DVD improved their readiness for screening and decreased screening barriers [27]. FGD participants in this study recommended receiving information about a postal intervention before its delivery, such as via banners in public places. According to a systematic review of controlled CRC screening trials, test completion rates ranged between 4–20% when an invitation attached to the FOBT kit was posted. Also, the provision of advanced notification consistently increased CRC screening uptake by 3–11% [28]. However, this systematic review did not include studies from Asian countries, and the behaviour of a population may vary depending on factors such as health literacy and social norms.

A common barrier appeared to be financial constraints related to accessing private healthcare. However, an iFOBT is free for Malaysians in public health clinics and there is a need to educate the population about their options. Participants seemed unclear about the difference between a screening test (iFOBT) and diagnostic test (colonoscopy). Public health clinics charge a nominal fee of RM1 to screen newly registered patients aged ≥50 years and there is no fee for older adults aged ≥60 years. Cancer patients in other studies refused screening due to a lack of financial resources to cover colonoscopy costs [29,30]. The anticipation of further costs might stop people from completing preventative screening tests. Findings from a large cross-sectional study with 2408 participants from eight different cities throughout Peninsular Malaysia [31] suggested that only 37% of participants were willing to pay for a colonoscopy and 47% of participants waited 6–8 months to receive a free colonoscopy from a government hospital. This finding highlights the problem of long waiting times. This finding points to a health inequity between people who have the means to access private health care, such as screening services available with short waiting times, and the majority of the population who depend on public health care with long waiting times and, ultimately, a delayed diagnosis.

The findings highlight low CRC awareness among Malaysian residents in Segamat, which is similar to previous quantitative findings i.e. 42% had poor knowledge about CRC symptoms, 50% had low awareness of CRC risk factors, and the availability of screening was unknown by 65.2% [32]. Awareness of the importance of CRC screening is crucial for CRC screening participation [33]. These combined qualitative and quantitative findings contribute to our understanding of the reasons for the low uptake of CRC screening in Malaysia [34]. The findings also suggest that doctors are crucial in encouraging the community to participate in preventative and CRC screening. Doctors were the primary source of information for FGD participants and highly influenced their health decision-making, apart from a few who self-medicated and refused to visit a doctor. This finding is in line with a nationwide survey about the use of medicines where approximately 84% of the Malaysian population reported that they would consult a physician for any health problem [35]. Participants in this study stated their willingness to attend follow-up visits for monitoring and managing health conditions (e.g. hypertension and diabetes) at public healthcare facilities. However, the findings appeared to indicate that participants' experience, even when they visited their local clinic, did not usually tend to include a discussion or recommendation about CRC screening. Similar findings have been reported from two regional surveys [36,37], and studies from other countries, such as the USA, that suggested that doctors may provide insufficient prompting and guidance, which, in turn, hinders CRC screening uptake [38,39]. Doctors play a crucial role in patients' decisions about illness prevention and health promotion behaviours, such as CRC screening [40]. There may be a need to help doctors fulfil this role and encourage them to be more active in identifying and informing patients who meet eligibility criteria for screening. Doctors also need to emphasise the importance of asymptomatic screening and provide government-funded cancer education programs for the target population.

Raising awareness about CRC and screening is an essential aspect of CRC prevention. The findings from this study, like several studies in Malaysia, highlighted low public awareness about CRC, symptoms, and screening [41]. There is a need to promote CRC awareness and screening activities through, for example, health campaigns [42] and general practitioners (as noted above). The use of incentives (e.g. payment of transportation fees, gift packs and food) may further enhance the uptake [43]. A positive finding was that participants understood the benefits of screening without any indication of symptoms and were willing to participate in CRC screening programs. Participants noted some specific aids to screening uptake, such as telephone or text message reminders to complete and return a sample, face-to-face communication of iFOBT positive results and relaying iFOBT negative effects through a telephone call or letter.

## Strengths and limitations

This study provides a clear picture of facilitating factors and barriers for participating in CRC screening from the perspective of the target population, including the range of different ethnic communities in Malaysia. It is the first study to explore the challenges and acceptability of delivering a stool collection container through postal services in Malaysia. It is important to note that participants were recruited from one state and findings may not apply directly to rural communities in other parts of Malaysia though studies elsewhere in Malaysia have reported similar barriers to CRC screening [42]. Overall, the study presents views of a range of members of the target population, across various age, sex, and ethnic groups. The study focused on iFOBT screening only, and other screening modes with other populations require separate attention.

## Conclusion

Overall, CRC is still relatively unknown amongst Malaysians despite its increasing incidence and late diagnosis. Common barriers to screening were waiting times at clinics, lack of transport, emotional barriers, and financial concerns. Culturally appropriate screening activities that consider the most commonly perceived barriers to iFOBT screening are required, including written and illustrated health educational materials and Malay, Chinese and Tamil reminders. The reported absence of discussion or recommendation from doctors undertaking screening appeared to be a barrier to the uptake of iFOBT screening, and there is a need for further research to investigate adherence by doctors to clinical guidelines about CRC screening in the ethnically diverse population of Malaysia.

## Supporting information

**S1 Checklist.**
(DOCX)

**S1 Appendix. Interview guide.**
(DOCX)

**S2 Appendix. Themes and quotes.**
(DOCX)

## Acknowledgments

The authors would like to thank the community members from Segamat, Johor, for their support and participation in this study.

## Author Contributions

**Conceptualization:** Désirée Schliemann, Michael Donnelly, Tin Tin Su.

**Funding acquisition:** Michael Donnelly, Tin Tin Su.

**Investigation:** Désirée Schliemann.

**Methodology:** Kogila Ramanathan, Désirée Schliemann, Michael Donnelly, Tin Tin Su.

**Project administration:** Kogila Ramanathan, Désirée Schliemann.

**Resources:** Kogila Ramanathan, Désirée Schliemann.

**Software:** Kogila Ramanathan, Désirée Schliemann.

**Supervision:** Désirée Schliemann, Nor Saleha Binti Ibrahim Tamin, Devi Mohan, Michael Donnelly, Tin Tin Su.

**Validation:** Désirée Schliemann, Devi Mohan, Michael Donnelly, Tin Tin Su.

**Visualization:** Michael Donnelly, Tin Tin Su.

**Writing – original draft:** Kogila Ramanathan.

**Writing – review & editing:** Kogila Ramanathan, Désirée Schliemann, Nor Saleha Binti Ibrahim Tamin, Devi Mohan, Michael Donnelly, Tin Tin Su.

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
