## [Decision Letter · Decision Letter 0]

16 Aug 2022

PONE-D-22-18410Facilitators and barriers to colorectal cancer screening in Malaysia: a qualitative studyPLOS ONE

Dear Dr. Su,

Thank you for submitting your manuscript to PLOS ONE. After careful consideration, we feel that it has merit but does not fully meet PLOS ONE’s publication criteria as it currently stands. Therefore, we invite you to submit a revised version of the manuscript that addresses the points raised during the review process.

We look forward to receiving your revised manuscript.

Kind regards,

Sanjiv Mahadeva, MRCP, MD

Academic Editor

PLOS ONE

Journal Requirements:

Additional Editor Comments:

Well conducted qualitative study. Will need to address comments by reviewers

Reviewers' comments:

Reviewer's Responses to Questions

**Comments to the Author**

1. Is the manuscript technically sound, and do the data support the conclusions?

Reviewer #1: Partly

Reviewer #2: Partly

2. Has the statistical analysis been performed appropriately and rigorously? 

Reviewer #1: No

Reviewer #2: N/A

3. Have the authors made all data underlying the findings in their manuscript fully available?

Reviewer #1: No

Reviewer #2: Yes

4. Is the manuscript presented in an intelligible fashion and written in standard English?

Reviewer #1: No

Reviewer #2: Yes

5. Review Comments to the Author

Reviewer #1: This paper presents facilitators and barriers to colorectal cancer screening in Malaysia. I greatly appreciate the time and efforts the authors put into it. However, several major issues should be addressed before it is published:

1. The study title is slightly misleading, as the exploration of barriers and facilitators is limited to stool-based screening test (iFOBT), which is recommended only for average-risk individuals in Malaysia. High-risk individuals, such as those with a family history of cancer, are not covered in this study.

2. The abstract, particularly the result section, is hard to follow, as all the factors are jumbled up. A clear key message is absent.

3. The description of the CRC program in Malaysia is inappropriate and likely to be incomprehensible for the readers who are not familiar with it - What do you mean by “no compulsory registration” and “identifying people and inviting them”?

4. The study objective is not specific and confusing. Terms like “enables” and “benefits” could carry the same meaning.

5. While this is supposed to be a qualitative study, it is described as a survey under the methodology.

6. The authors claim that the discussion guide was designed based on the relevant literature, but only one reference is cited. This raises a concern if a thorough literature review was performed. Was any framework adopted for the thematic analysis?

7. Why must the authors emphasize “a female researcher”? Were there any gender-sensitive issues in the FGD? If so, it is questionable that how appropriate it was to conduct an FGD mixing male and female participants.

8. It is hard to visualize how participants speaking different languages could be gathered with the assistance of translators for an FGD, which is meant to be interactive.

9. How could identifiable information be technically removed from audio recordings? Why was this necessary?

10. Were the transcripts returned to FGD participants for validation?

11. How did the authors confirm data saturation?

12. It is inappropriate to conclude that “certain groups tended to ….”, as no quantitative analysis was performed.

13. Each point under a theme should be supported only by one quote to avoid redundancy.

14. The points under each theme seemed to be disjointed.

15. Some of the themes did not even have a direct relationship with CRC screening.

16. The references cited in the discussion are not directly related to the findings in most cases.

17. Grammatical errors are present throughout the paper. An edit for succinctness and brevity is also required.

Reviewer #2: Thank you for inviting me to review this manuscript.

The authors have reported on a topic which is highly relevant and important for bowel cancer control. This qualitative study has uncovered several critical issues that may require further attention in order to encourage uptake of bowel screening amongst the Malaysian population in the future.

There are several areas in the manuscript which require further clarification:

1. Sampling

- Purposive sampling was conducted for this study and this was aided by a community organization called SEACO. Some inclusion criteria were mentioned but were there exclusion criteria – would prior personal experience of bowel screening affect recruitment into the study. What measures were taken to minimise sampling bias?

- The cohort of individuals recruited into the study are primarily from a rural community and where most of the interviewees’ educational level are up to secondary education. I feel the word rural community needed to be included in the title as these findings may not be truly representative of Malaysia as a whole nation. This should be mentioned in the limitation’s chapter.

2. Methodology

- During the FGD, were attempts made to reduce bias from dominance effect, halo effect and group thinking?

- The data analysis was described in phases but could do with further elaboration. The authors used an iterative approach but what method was used to move from codes to themes (e.g. inductive? Deductive analysis?).

- There was no mention of attempts made to determine whether data saturation was achieved?

3. Results

- The results section was mainly descriptive and significant sections lacked in-depth analysis. For example lines 208 -228: Cancer awareness theme, section of Barriers to CRC screening. Statements were made to summarise the included quotes but there was limited analysis of how these subthemes may relate and influence decisions to screening. The theme on Barriers to CRC screening is also rather short – particularly on stool collection - this is probably one of the most important subthemes that has been uncovered and warrants further elaboration and specifics.

- There were statements which did not fully reflect the included quotes e.g. line 247. The statement about participants’ perception of self susceptibility to CRC is not supported by the included transcript.

- Under Theme 5 – suggested strategies to improve iFOBT uptake or completion, are the authors implying that certain strategies are more likely to be suited for specific ethnic groups as several statements have attempted to draw associations with certain ethnic groups?

4. Conclusion

Line 39 – ‘Another barrier that people faced was financial constraints’.

Whilst I can see what the authors are alluding to, the data could also be interpretated as bowel screening is affordable to everyone through the public health system in Malaysia, but this is limited by capacity – long waiting times etc. An alternative would be to seek private health care screening facilities, but this would incur costs to the patient and would thus limit access to those who cannot afford it. I would therefore recommend clarifying the first sentence to this paragraph.

5. Others

- Is there a typo on Line 106 – 3rd section mentioned but 2nd section was not mentioned

- Typo on Line 180? – ‘stated that they would not ask for other preventative health screenning’ – is this true given the subtheme here is about participants reporting that they would actively ask their doctor about cancer screening

- Line 358 – missing or incomplete sentence

6. PLOS authors have the option to publish the peer review history of their article (what does this mean?). If published, this will include your full peer review and any attached files.

Reviewer #1: No

Reviewer #2: No

---

## [Author Response · Author response to Decision Letter 0]

1 Nov 2022

Dear PLOS ONE editorial team,

Thank you very much for providing us with the opportunity to revise the manuscript. We would also like to thank the reviewers for providing feedback and comments to help us improve the manuscript. We have addressed the comments as outlined below and highlighted in the manuscript. 

 Journal Requirements:

Reply: We have relabelled the files as requested. 

Reply: We have updated the information as requested. 

Reviewers' comments:

Reviewer #1:

This paper presents facilitators and barriers to colorectal cancer screening in Malaysia. I greatly appreciate the time and efforts the authors put into it. However, several major issues should be addressed before it is published:

1. The study title is slightly misleading, as the exploration of barriers and facilitators is limited to stool-based screening test (iFOBT), which is recommended only for average-risk individuals in Malaysia. High-risk individuals, such as those with a family history of cancer, are not covered in this study. 

Reply: Thank you for taking the time to read and provide feedback on this manuscript. We have changed the title of the manuscript accordingly to highlight that this study is focussing on the iFOBT. (Line 1-3)

2. The abstract, particularly the result section, is hard to follow, as all the factors are jumbled up. A clear key message is absent. 

Reply: We have revised the results section. (Line 33-39)

3. The description of the CRC program in Malaysia is inappropriate and likely to be incomprehensible for the readers who are not familiar with it - What do you mean by “no compulsory registration” and “identifying people and inviting them”? 

Reply: We have revised this section. (Line 69 -75)

4. The study objective is not specific and confusing. Terms like “enables” and “benefits” could carry the same meaning. 

Reply: We have revised the objective to be clearer. (Line 25-28 and 77-80)

5. While this is supposed to be a qualitative study, it is described as a survey under the methodology.

Reply: Thank you for your comment. We followed the COREQ guidelines to describe the methods section and repeatedly refer to focus group discussion as a way of interviewing participants. Please do let us know any specific parts that you feel should be changed. 

6. The authors claim that the discussion guide was designed based on the relevant literature, but only one reference is cited. This raises a concern if a thorough literature review was performed. Was any framework adopted for the thematic analysis?

Reply: We have updated the information in the methods section. (Line 115-117 and 141-149)

7. Why must the authors emphasize “a female researcher”? Were there any gender-sensitive issues in the FGD? If so, it is questionable that how appropriate it was to conduct an FGD mixing male and female participants.

Reply: There were no gender-sensitive issues in the FGD. However, the pilot FGD was conducted with a group of both male and female participants and we noticed that some female participants felt shy and did not participate actively in the discussion. Feedback then suggested that they would be more comfortable participating in the FGD with other women only. Therefore, we planned separated the FGDs by gender. Men felt comfortable talking to either gender. Both male and female participants were comfortable with the female facilitator and researcher. (Line 130-134)

8. It is hard to visualize how participants speaking different languages could be gathered with the assistance of translators for an FGD, which is meant to be interactive.

Reply: Malay is an official and main communication language in Malaysia. Most Malaysian citizens can communicate using basic Malay. However, some older participants in Chinese ethnic groups have low confidence in their fluency in Malay to discuss health-related issue. The interviewer could not speak Chinese dialects, therefore we needed a translator for the focus group discussion with Chinese. The role of the translator to not to translate words by words but for some complicated discussion. The interviewer asked questions in Malay and the translator translated them into Mandarin for the participants and then translated the Mandarin response into Malay for the other participants and interviewer. It was an interactive session because a few of the participants were able to speak in Malay. Those who could not communicate in Malay could still discuss in Mandarin.

9. How could identifiable information be technically removed from audio recordings? Why was this necessary?

Reply: We revised this as follows in the manuscript and hope that this is clear. The interviews were recorded using voice recorders. The audio recordings were transcribed, and any identifiable information (i.e. name) was removed from the transcript. (Line 137-138)

10. Were the transcripts returned to FGD participants for validation? 

Reply: No. We checked the quality of the transcripts for accuracy after they were transcribed. The findings were presented during Community Engagement Committee’s annual meeting which was held online on the 31st March 2022.

11. How did the authors confirm data saturation?

Reply: Originally we planned 2 FGDs with each ethnic group. However, we conducted three more FGDs with selected ethnicities because the number of participants was below six for some of the FGDs. By the third FGD, it was clear that participants faced similar challenges and which was confirmed by the analysis. (Line 150- 154)

12. It is inappropriate to conclude that “certain groups tended to ….”, as no quantitative analysis was performed.

Reply: We changed this accordingly. Thank you. 

13. Each point under a theme should be supported only by one quote to avoid redundancy.

Reply: We included quotes that we thought are most suitable to represent each theme. Some themes have 2 quotes to better show the differences and similarities expressed by different ethnic groups and genders. 

14. The points under each theme seemed to be disjointed.

Reply: We believe that the sub-themes fit well under each key-theme. These are further explained in the results section which we hope offers explanations as to how they are linked. If there is anything in specific that is unclear or seems disjointed, please let us know. 

15. Some of the themes did not even have a direct relationship with CRC screening.

Reply: Some of the themes have an indirect relationship with CRC screening. As described in the methods, we first asked general questions about health-seeking behaviour and accessing prevention as the literature suggests that attitudes towards health-seeking in general can be a good indicators as to whether or not people will take up screening when this is offered. We believe that the results eg general trust in doctor are important and informative for future cancer prevention and CRC screening research. 

16. The references cited in the discussion are not directly related to the findings in most cases 

Reply: There were few studies that conducted qualitative research with regards to CRC screening in the past. All the references provided are related to CRC screening, cancer and health-seeking. If there is anything that stands out in particular as unrelated, please let us know. 

17. Grammatical errors are present throughout the paper. An edit for succinctness and brevity is also required.

Reply: We edited the paper for grammar. Thank you very much for your comments. 

Reviewer #2: 

Thank you for inviting me to review this manuscript.

The authors have reported on a topic which is highly relevant and important for bowel cancer control. This qualitative study has uncovered several critical issues that may require further attention in order to encourage uptake of bowel screening amongst the Malaysian population in the future.

There are several areas in the manuscript which require further clarification:

1. Sampling

- Purposive sampling was conducted for this study and this was aided by a community organization called SEACO. Some inclusion criteria were mentioned but were there exclusion criteria – would prior personal experience of bowel screening affect recruitment into the study. What measures were taken to minimise sampling bias?

Reply: Thank you very much for taking the time to review this manuscript. Regarding the exclusion criteria, we excluded those aged below 50 years of age, participants who were not able to provide responses independently and had a history of colorectal cancer. (Line 100-102)

To reduce sampling bias, we contacted the Community Engagement Committee which includes representatives from respective communities who are knowledgeable about the community. We had representatives from 5 sub-districts and we had FGD sessions including participants from these areas. The Project leader and researcher (KR) contacted eligible community members who were put forward by the CEC to explain the study. We ensured that participants represented different age groups, ethnic groups, genders as well as working and retired participants.

-The cohort of individuals recruited into the study are primarily from a rural community and where most of the interviewees’ educational level are up to secondary education. I feel the word rural community needed to be included in the title as these findings may not be truly representative of Malaysia as a whole nation. This should be mentioned in the limitation’s chapter.

Reply: We changed the title to semi-rural as not all areas included in Segamat fit the definition of rural (Line 1-3). The education attainment (i.e. who completed secondary level) among Malaysians is 62.9% in 2019 (source: world bank data available from https://data.worldbank.org/indicator/SE.SEC.CUAT.UP.ZS?locations=MY Therefore, it is similar to study participants' education level as well. 

2. Methodology

- During the FGD, were attempts made to reduce bias from dominance effect, halo effect and group thinking?

Reply: Yes, the interviewer encouraged responses from all participants and allowed pauses throughout the discussion to allow everyone to voice their opinion and avoid a dominance effect. The responses were from all the participants, and the researcher did not make assumptions or generalizations during the discussion. 

- The data analysis was described in phases but could do with further elaboration. The authors used an iterative approach but what method was used to move from codes to themes (e.g. inductive? Deductive analysis?).

Reply: We added more detail in the methods section. (Line 141-149)

- There was no mention of attempts made to determine whether data saturation was achieved?

Reply: Originally we had planned 2 FGDs with each ethnic group. However, we conducted three more FGDs with selected ethnicities because the number of participants was below six for some of the FGDs. By the third FGD, it was clear that participants faced similar challenges and which was confirmed by the analysis. (Line 150- 154)

3. Results

- The results section was mainly descriptive and significant sections lacked in-depth analysis. For example lines 208 -228: Cancer awareness theme, section of Barriers to CRC screening. Statements were made to summarise the included quotes but there was limited analysis of how these subthemes may relate and influence decisions to screening. 

Reply: We added more information under the cancer awareness theme (Line 206-208, 214-215, and 227) 

-The theme on Barriers to CRC screening is also rather short – particularly on stool collection - this is probably one of the most important subthemes that has been uncovered and warrants further elaboration and specifics.

Reply: We elaborated further on embarrassment for stool collection.(262-265) 

- There were statements which did not fully reflect the included quotes e.g. line 247. The statement about participants’ perception of self-susceptibility to CRC is not supported by the included transcript.

Reply: We changed this accordingly. (Line 241-242) 

- Under Theme 5 – suggested strategies to improve iFOBT uptake or completion, are the authors implying that certain strategies are more likely to be suited for specific ethnic groups as several statements have attempted to draw associations with certain ethnic groups?

Reply: We indicated that collaboration with the community support organisations, that are active in their area, to deliver screening programmes would improve CRC awareness and support screening activities for successful engagement with the community. 

4. Conclusion

Line 39 – ‘Another barrier that people faced was financial constraints’.

Whilst I can see what the authors are alluding to, the data could also be interpretated as bowel screening is affordable to everyone through the public health system in Malaysia, but this is limited by capacity – long waiting times etc. An alternative would be to seek private health care screening facilities, but this would incur costs to the patient and would thus limit access to those who cannot afford it. I would therefore recommend clarifying the first sentence to this paragraph.

Reply: We revised the first sentence of the paragraph. (Line 375-376)

5. Others

- Is there a typo on Line 106 – 3rd section mentioned but 2nd section was not mentioned

Reply: We corrected this. (Line 119-120) 

- Typo on Line 180? – ‘stated that they would not ask for other preventative health screening’ – is this true given the subtheme here is about participants reporting that they would actively ask their doctor about cancer screening

Reply: We revised this. (Line 182-185) 

- Line 358 – missing or incomplete sentence

Reply: We revised this. (Line 336-337)

---

## [Decision Letter · Decision Letter 1]

9 Dec 2022

Facilitators and barriers to colorectal cancer screening using the immunochemical faecal occult blood test among an average-risk population in semi-rural Malaysia: a qualitative study

PONE-D-22-18410R1

Dear Dr. Su,

We’re pleased to inform you that your manuscript has been judged scientifically suitable for publication and will be formally accepted for publication once it meets all outstanding technical requirements.

Kind regards,

Sanjiv Mahadeva, MRCP, MD

Academic Editor

PLOS ONE

Additional Editor Comments (optional):

The revised manuscript is much improved. This paper is of importance within this area of research

Reviewers' comments:

Reviewer's Responses to Questions

**Comments to the Author**

1. If the authors have adequately addressed your comments raised in a previous round of review and you feel that this manuscript is now acceptable for publication, you may indicate that here to bypass the “Comments to the Author” section, enter your conflict of interest statement in the “Confidential to Editor” section, and submit your "Accept" recommendation.

Reviewer #2: All comments have been addressed

2. Is the manuscript technically sound, and do the data support the conclusions?

Reviewer #2: Yes

3. Has the statistical analysis been performed appropriately and rigorously? 

Reviewer #2: N/A

4. Have the authors made all data underlying the findings in their manuscript fully available?

Reviewer #2: Yes

5. Is the manuscript presented in an intelligible fashion and written in standard English?

Reviewer #2: Yes

6. Review Comments to the Author

Reviewer #2: My comments from the previous review have been adequately addressed in this latest draft. Recommend acceptance of the manuscript.

7. PLOS authors have the option to publish the peer review history of their article (what does this mean?). If published, this will include your full peer review and any attached files.

Reviewer #2: No

---

## [Editor Report · Acceptance letter]

19 Dec 2022

PONE-D-22-18410R1 

*Facilitators and barriers to colorectal cancer screening using the immunochemical faecal occult blood test among an average-risk population in semi-rural Malaysia: a qualitative study*

Dear Dr. Su:

I'm pleased to inform you that your manuscript has been deemed suitable for publication in PLOS ONE. Congratulations! Your manuscript is now with our production department. 

Kind regards, 

on behalf of

Prof Sanjiv Mahadeva 

Academic Editor

PLOS ONE